# Quality of Life and Stress Management in Healthcare Professionals of a Dental Care Setting at a Teaching Hospital in Rome: Results of a Randomized Controlled Clinical Trial

**DOI:** 10.3390/ijerph192113788

**Published:** 2022-10-23

**Authors:** Fabrizio Guerra, Denise Corridore, Margherita Peruzzo, Barbara Dorelli, Lucrezia Raimondi, Artnora Ndokaj, Marta Mazur, Livia Ottolenghi, Giuseppe La Torre, Antonella Polimeni

**Affiliations:** 1Department of Oral and Maxillofacial Sciences, Sapienza University of Rome, 00185 Rome, Italy; 2Department of Public Health and Infectious Diseases, Sapienza University of Rome, 00185 Rome, Italy; 3University of Tor Vergata, 00133 Rome, Italy

**Keywords:** quality of life, stress management, healthcare professionals, dentists, care for carers, medicine, randomized controlled clinical trial, yoga

## Abstract

In the healthcare environment, more and more people experience work-related stress. The Faculty of Medicine and Dentistry of Sapienza University of Rome, having observed the need to take care of healthcare professionals, has set itself the objective of providing useful tools such as additional and necessary personal protective equipment for healthcare professionals. Objective: To promote health through better management, skills, and the use of strategies and solutions to identify, decompress, and neutralize those responsible for work-related stress mechanisms in order to take care of those who care (CURARE CURANTES). Materials and Methods: A randomized controlled clinical trial was conducted. The participants were enrolled by three departments of the Department of Dental and Maxillofacial Sciences of Rome; in consideration of emotional involvement of health professionals who work there. A motivational program was offered. Results: There were 17 and 16 healthcare professionals in the intervention and control groups, respectively. Levels of the mental composite score (MCS) varied both in the pre–post phase of the intervention group (*p* = 0.002), and between the intervention group and control group in the post phase (*p* = 0.006). No significant differences were observed for the physical composite score (PCS). Similarly, there were no significant differences regarding the positivity scale (PS) and the two dimensions of work-related stress (job demand and decision latitude). Conclusion: This study demonstrates the efficacy of yoga training practiced directly in the workplace and wearing work clothes, entering the work context, not weighing further on the healthcare workload, and being a way to carry out physical activity even in those cases in which professionals do not have the time to do it in their free time.

## 1. Introduction

The Faculty of Medicine and Dentistry of Sapienza—University of Rome—observes the need to care for healthcare professionals through the implementation of aid and support programs. It has been working to create fields of action in favor of healthcare professionals for years, making available specific techniques selected from the vast “yoga world and oriental health techniques” fully adapted to the work environment and effective at contributing to the wellbeing of healthcare professionals, and therefore also of the Local Healthcare Authority, to counteract the problems of the healthcare profession in this context and promote the health of its workers [1,2,3,4]. Considering the continuous emergency experienced in the health sector and its exacerbation resulting from the pandemic and its ongoing consequences, the fundamental need to support the wellbeing of health professionals has also been recognized. The “CURARE CURANTES” project is part of this plan, aimed at promoting the health and wellbeing of health personnel, through accessible and functional methods deriving from the selection and development of health techniques, both western and eastern. Due to several factors, the healthcare environment can be a cause of stress, and more and more burnout syndromes are manifested in dealing with problems related to the profession on a daily basis [5,6,7]. The relationship with patients, in addition to the provision of care, requires the ability to contain and take charge of the emotional impact that accompanies the experience of illness and pain in the patient [8]. “Difficult” cases, the management of certain diseases, emergencies, shifts, the work environment, the relationship with colleagues, administrative problems, and legal problems deeply involve healthcare professionals, not only as a manifestation of physical fatigue, but also as mental unease and an inability to manage emotional stress [3,9,10,11,12,13,14,15].

This increases our motivation to evaluate the effectiveness of the techniques, with the aim of making them available to the entire health, scientific, and administrative community, as well as to students of the health professions [16,17].

These elements are particularly highlighted by some categories of professionals such as dentists, who are exposed to severe stress in their particularly challenging work area [18,19].

It is well known that Penfield’s sensory homunculus represents the somatosensory cortex involved in tactile sensitivity, temperature, and pain. Sensory information arrives in this area to be processed and allow us to perceive the world and body sensations. The mouth is one of the most sensitive areas, characterized by a greater number of receptors, and is larger in the design of the homunculus [20]. Carrying out invasive therapies on such a particularly sensitive area, and invading the personal space of the patient under conditions of full vigilance and maximum capacity, increases the stress of the dentist. The rise of the “consumer” health service is particularly relevant to dentistry, where most care is (at least in part) paid for, and is likely to fuel the stress to which the practitioner is subjected [21].

Scholars understand how a reciprocal exchange between professionals through a transformative, experiential activity that generates an initiative in the work environment leads to significant changes in favor of dental professionals. There is evidence that dental practitioners during the course of their professional activity are exposed to several occupational hazards, including physical, chemical, biological, and ergonomic factors. Considering the last hazard, associated with strained posture and prolonged repetitive movements, can induce musculoskeletal disorders [22]. From the systematic review of the literature conducted by Cocchiara RA et al. [1], it emerges, in fact, how many stress management techniques are effective in the prevention and management of musculoskeletal and psychological problems, leading to an improvement in physical problems and sleep quality, and reducing stress levels, as has also been shown in recent studies [3,23]. The study conducted by Chismark A et al. [11] shows how complementary and alternative medicine (CAM) therapies can improve the quality of life, reduce work interruptions, and increase professional satisfaction for dental hygienists suffering from chronic musculoskeletal pain.

This randomized controlled clinical trial aimed to evaluate changes in the quality of life and work-related stress in a group of health professionals operating in a dental setting in the context of the COSMOS project, included in the “CURARE CURANTES” project.

## 2. Materials and Methods

### 2.1. Setting

The participants were enrolled in the autumn of 2019 from three departments of the Umberto I Clinic in Rome: the MoMax Department (Oral and Maxillofacial Medicine), the Department of Paediatric/Special Needs Dentistry, and the First Visit and Registration Department of the Dental Section of the Head and Neck Integrated Department. The project was conducted in consideration of the highly complex clinical cases and the subsequent psychological/physical and emotional involvement of the health workers working at the clinic. Before the intervention, the purpose of the study was explained and, after a verbal consent to participate, a written consent form based on the Declaration of Helsinki was sent via email, asking participants to return it completed and signed (Ethics Committee of Department of Oral and Maxillofacial Sciences, Sapienza University of Rome N. 54/2019 Prot. n. 0001269 del 24/07/2019).

### 2.2. Sample Sizing

Sample sizing was calculated using the following parameters:Alpha level = 5%.Study Power = 80%.A 10% increase in MCS in the intervention group.

Based on these parameters, it was estimated that 32 health professionals (16 per group) had to be enrolled. Considering a drop-out level of 20%, in the end, 40 individuals were enrolled, divided into two groups, the study group and the control group, with a total of 20 individuals per group.

Randomization took place using random numbers electronically generated using the software EpiCalc 2000. Different lists were used for different genders.

### 2.3. Inclusion and Exclusion Criteria

The following criteria were considered.

Inclusion criteria: age group between 25 and 65 years; working in the selected departments.

Exclusion criteria: workers serving as administrative staff.

### 2.4. The Intervention

To promote the wellbeing of the healthcare professionals, preventing and compensating for any unease in the work environment and avoiding the limitations that could occur, a range of practical exercises was offered, suitably chosen for this work environment, as opportunities to seek a shared sense of balance.

The method used derives from the development of techniques from yoga, meditation, martial arts, and training in empathic communication, to confer the ability to be present and develop the power to change or modify one’s state of mind, favoring an optimal response to the work required with: clarity, self-control, resistance, optimal detachment, focus, psychological/emotional balance, ability to recover quickly from fatigue [9,10,24,25]. The selection included dynamic exercises, positions, breathing techniques, and meditations practiced in India for millennia. The selection criterion was based on the immediate effectiveness of these techniques given the tight deadlines available to a health professional, centered on the solution to work-related stress, and indicated to create immediate benefits.

Short training sets lasting 15 min were performed on specific points, i.e., motivation, movements, breathing, and relaxation, and presented as a simple “secular” tool, in the absence of the ideological and theological component typical of these disciplines.

Physical training implements respiratory, mental, and behavioral skills, and teaches users to master the modulation of states of consciousness in order to influence body processes towards greater health, wellbeing, and better psychophysiological conditions [2,26,27].

A place was chosen in the clinic facility for the pre-set meetings and indications were given on the type of clothing to be worn; given the comfort of the uniform used, it was considered appropriate to use it as clothing suitable for the proposed work. By mastering these simple techniques, you can use them in the ward, without the need for any specific setting, according to the work environment.

The intervention took place in 15-minute appointments performed twice a week for four weeks and implemented as two appointments per day: Before the shift to encourage motivation and support for professionals involved in department activities to boost strength, determination, tolerance, good mood, efficiency, and balanced compassion.At the end of the shift, with the provision of another 15 min to allow quick decompression, facilitate relaxation, and its beneficial effect in favor of rebalancing, and releasing anxiety and stress caused by the emotional conditions in which they have been immersed during work to facilitate physiological rest and optimal conditions of the psychological/physical and emotional systems.

The following scheme was followed:(a)Brief verbal portion aimed at motivation.(b)Performance of various exercises, stretching and relaxation exercises of various parts of the body; muscular, visceral, and connective tissue. Work focused mainly on the shoulders, neck, chest, back, pelvis, and hamstrings (areas that are more likely to maintain tension).(c)Breathing techniques (Pranayama) performed simultaneously with physical exercises, slowing down and consciously channeling the breath to counteract a series of physiological components of stress while reducing feelings of anxiety.
○Increase in the length of exhalation compared to inhalation ○Fractional breathing ○Relaxed diaphragmatic breathing training. Main techniques carried out:
Accelerated thoracic and diaphragmatic rhythmic breathing (Bhastrika),Alternating nostril breaths (Nadishodana),Fast breathing technique that involves the stomach and belly (Kapalabhati)(d)Specific meditations, movements, focalizations, and positions to free oneself from the tension that remains trapped in the body, help balancing emotions, recovering from fatigue, and reaching a state of mental neutrality.(e)Use of sound repeated to aid concentration and calm the restless mind (mantra).(f)Music mixed with the performance of exercises and meditations and for relaxation to inhibit thought production and facilitate inner calm and promote a physiological breathing rhythm.(g)Relaxation sessions.

Particular attention was paid to this final part, since, in a state of deep relaxation, a change occurs in favor of physiological balance, as it is the optimal environment for cell function [28].

Each session closed with the adoption of relaxation positions on the ground on the back (Shavasana), as well as changes in the position of the legs to open the hip, shoulder, and thoracic vertebral segment for an increase in the relaxation space, combined with relaxation guided by the teacher’s voice (Nidra). Techniques to enhance deep relaxation and calm the hypervigilance associated with fatigue towards a dreamlike state that facilitates energy recovery [28,29].

Rapid techniques have also been suggested to be implemented during working hours to maintain, manage, and increase focus, lucidity, and breath control.

### 2.5. The Questionnaires

Before the intervention at time T0, online questionnaires were administered to both selected groups, as well as at the end of the intervention.

The following questionnaires were used:-The Italian version of SF12 for the assessment of quality of life, which allowed for the calculation of the mental composite score (MCS) and physical composite score (PCS) indicators [30].-Positivity scale for the assessment of positivity [31].-The Italian version of the 15-item Karasek questionnaire for the assessment of the two dimensions of work-related stress; job demand and decision latitude [32].

#### 2.5.1. SF-12

This questionnaire was developed as a shorter form of the SF-36 (Short Form Health Survey). SF-36 is a 36-item survey of patient health. It was created by the Medical Outcome Study (MOS) [33], and is designed for use in clinical practice and research, health policy evaluations, and general population surveys. Its purpose is to provide a short but reliable survey directly to the individual and assess their health status, evaluating eight different health components. The SF-12, an even shorter survey, was subsequently developed and adapted to assess a person’s state of health. It allows the researcher to investigate mental and physical health through two different summary scores: MCS-12 and PCS-12, respectively.

#### 2.5.2. Positivity Scale

The eight-item positivity scale (PS) was used to measure positivity, defined as the tendency to view life and experiences from a positive perspective. Participants responded to eight points on a five-point scale from 1 (strongly disagree) to 5 (strongly agree).

#### 2.5.3. Robert Karasek Questionnaire

The 15-question version of the Karasek questionnaire was administrated according to the demand/control approach. This allowed us to calculate the two dimensions job demand and decision latitude.

The most used hypothesis for the demand/control model assumes that the most adverse psychological stress reactions occur when psychological demands are high and the worker’s decision latitude is low, i.e., work-related stress.

### 2.6. Statistical Analyses

The statistical analysis included the use of Mann–Whitney and Wilcoxon signed rank tests to assess the differences between groups and within groups, respectively. The analysis was conducted using SPSS 25.0 software.

Statistical significance was set at *p* < 0.05.

## 3. Results

All health professionals operating in the three selected departments were enrolled, consisting of 40 (Figure 1) divided into two groups, the study group, and the control group. Of these, three participants in the intervention group (of which one did not respond to the final questionnaire and two did not complete the planned intervention) and four participants in the control group (all of which did not complete the final questionnaire) were excluded from the analysis.

Table 1 shows the sociodemographic characteristics of the intervention group and the control group.

The two groups were comparable in terms of gender, age, marital status, parenting, and educational level.

Table 2 shows the results of the outcome variables considered in this study in both intervention and control groups.

In the current study, the following Cronbach’s alpha scores were found:-SF-12: 0.837-Positivity: 0.891-Karasek: 0.815

It can be observed that the levels of the MCS vary both in the pre–post of the intervention group (*p* = 0.002), and between the intervention group and the control group in the post phase (*p* = 0.006).

For the other dimension of the quality of life PCS, no significant differences were observed both within the pre–post groups nor between the groups at the two points of assessment.

Similarly, to the situation regarding PCS, there were no significant differences regarding the positivity scale and the two dimensions of work-related stress (job demand and decision latitude).

## 4. Discussion

Due to various factors, the healthcare environment can be a source of great stress that impacts both the physical/skeletal and psychological/emotional system, precisely because of the responsibilities that this type of profession entails, and the target population with which it is confronted. Especially at this point in history, it is necessary to focus attention on safeguarding the wellbeing of health professionals, who are called upon to face the health emergency arising from the COVID-19 pandemic more than any other profession. Moreover, we can affirm that the trial conducted has shown to be effective in the short-term improvement of mental health, measured through the MCS scale derived from the SF-12 questionnaire, though less so for the other indicators considered. Moreover, the improvement of the state of mental health using yoga techniques and for stress decompression has proved effective and timely in numerous other studies. The clinical study by Li A. et al. [2] reports a statistically significant reduction in stress levels in a population of nurses, as also reported by the studies by Alexander et al. [10] and Kemper et al. [4]. The results of the literature review carried out by Bischoff L.L. et al. [34] suggest an effect of stress-reducing yoga interventions on the healthcare personnel examined. By contrast, regarding positivity and the level of work-related stress, the results show the indicators to remain largely stable (positivity, job demand, and decision latitude), while there are no effects on the PCS scale, relating to the level of physical health, which does not confirm what has emerged in the scientific literature. The study by Koneru S et al. [12] found that yoga was more effective than other forms of physical activity such as aerobics, fast walking, sports, etc., reporting that 89.5% of yoga practitioners were free of musculoskeletal pain, also comparing it with other studies in which only 78.3% were free of musculoskeletal pain. It must be said that this group of dentists practiced yoga (under the supervision of a qualified yoga teacher) or some other physical activity, such as fast walking, jogging, or aerobics, for a period of more than one year and a minimum of four times a week for at least one hour a day, unlike our study group that practiced yoga for a very limited period and with relatively little intensity. In addition, for the aspects related to positivity, job demand, and decision latitude, in order to obtain an improvement impact on the indicators, there is a need for organizational and management changes, which obviously cannot depend on the individual, and have perhaps a greater latency to be able to manifest themselves, as also demonstrated by the study conducted by La Torre et al. [3]. Satisfaction at work is an important factor to be observed and is based on the determination of various factors, such as hours worked, earnings, relationships with colleagues, and the environment in general. In the study by Slabsinskiene et al. [35], the relationship values between job satisfaction and work-related stress level were as strong and significant as those found in the studies by Mijakoski et al. [36] and Molina-Hernández et al. [37]. Of course, the assessment of these parameters can be carried out in different ways and represents the personal consideration of the subjects examined, which makes it difficult to compare them. The benefit to mental health and to the individual shows a clear and significant result, but the need to evaluate the organizational and managerial elements remains to be considered, which are not measurable using the short-term effects. The experience of work-related stress that depends on shift work, administrative problems, legal problems, and the great demand due to emergencies continues to occur every day, and remains a sore point that deeply affects healthcare professionals. The feedback collected, despite the lack of evidence in the indicators, was positive, with participants stating improved wellbeing and increased strength, which defined a turning point to address stress and promote a feeling of immediate wellbeing. We intend to continue the research including the evaluation of clinical parameters.

### 4.1. Limitations of Study

Despite the encouraging results, it is necessary to highlight some limitations related to this study. One problem was the limited sample size, which suffered a further limitation due to the loss of subjects during the intervention for reasons related to work shifts that did not coincide with the needs of the study. Some other participants did not complete the final questionnaires and were excluded from the study. Another limitation was the limited time of intervention that did not allow us to incentivize and implement the improvements that were initially expected. We also note the presence of selection bias because dentists of a single facility were enrolled, and these may differ in terms of cultural background, training, geographical location, and personal context from their colleagues, despite being employed in the same sector. For these reasons, the results obtained must be interpreted with caution.

### 4.2. Strengths and Future Directions

The “CURARE CURANTES” project, consistently with what has been said, is aimed at promoting the health and wellbeing of healthcare personnel, through accessible and functional methods. The project’s future objective is the extension of the audience to the technical/administrative staff and to the students of the degree courses in the health area. It can also be noted as a strong point that the physical activity carried out by the participants in the study was practiced directly in the workplace and wearing work clothes, entering the work context, not weighing further on the healthcare workload, and being a way to carry out physical activity even in those cases in which professionals do not have the time to do it in their free time.

## 5. Conclusions

In the scientific literature, it increasingly emerges that work-related stress is responsible for problems and illnesses. For the maintenance of the state of health, both physical and nervous or emotional, prevention and education in this regard, as well as methodologies, are an effective means to ensure the health and wellbeing beyond the physiological sphere, including psychological and social factors. It is essential to receive education on lifestyles that include taking the time to care for yourself, focusing on breathing, nutrition, and physical exercise as natural techniques essential to maintaining one’s state of health.

Working in hospital environments means facing administrative problems every day, difficult cases, both in terms of the diagnosis and management of certain diseases, the ongoing possibility of finding oneself in emergency situations, and other factors that play an important role in stress, especially the constant contact with patients who are suffering, deeply involves health professionals at an emotional level, in addition to the physical exertion. The consequence of prolonged high levels of stress affects the individual, and can also impact upon the hospital itself.

Through the programmed implementation of interventions such as the one proposed by the present study, the “CURARE CURANTES” project, given the encouraging results obtained, it is possible to lighten the burden of health professionals, reducing the levels of work-related stress and implementing the benefits on the physical response as well as on the level of job satisfaction, setting aside the necessary time.

Given the simplicity, effectiveness, and accessibility of these methods (which received many favorable responses from all the healthcare professionals who have had the opportunity to experiment with the sets offered during the training courses and clinical trials carried out), their implementation requires nothing more than motivation on the part of the management of the healthcare companies to whom the protocol is addressed, in response to the numerous data now present in the literature with respect to the need to prevent the emotional and physical exhaustion of healthcare staff. The goal in the long term is to improve the relational and communicative aspects, the relationship with colleagues and patients, the promotion of the personalization of care, constantly negotiating between technical–scientific skills and transversal skills. Training in managing emotions, the ability to lighten and decongest problems, favors psychophysical endurance and tolerance.

## Figures and Tables

**Figure 1 ijerph-19-13788-f001:**
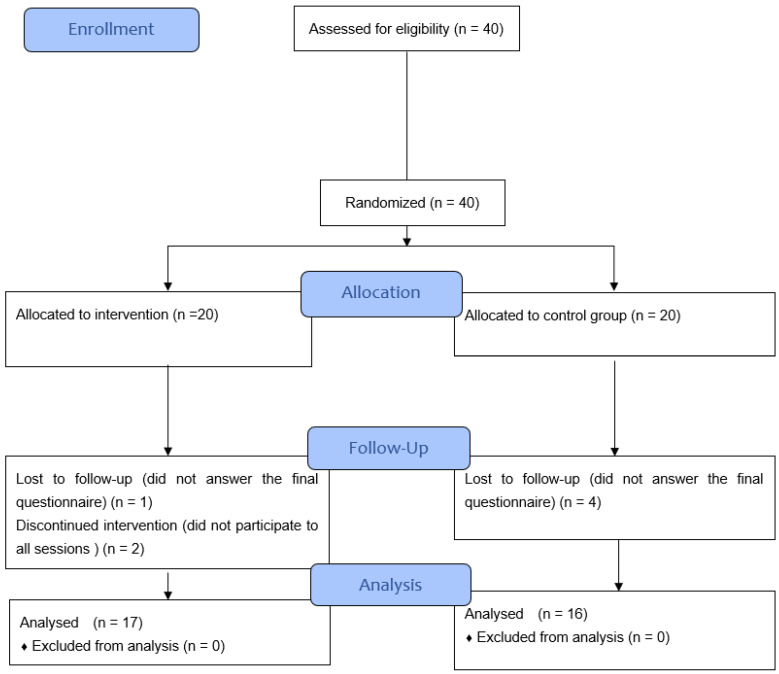
CONSORT 2010 flow diagram.

**Table 1 ijerph-19-13788-t001:** Characteristics of the two groups of participants to the trial.

Variables	Intervention Group N (%) or Mean (SD)	Control Group N (%) or Mean (SD)	*p*
*Gender*			0.611
Females	7 (41.2)	8 (50)	
Males	10 (58.8)	8 (50)	
*Age*	36.8 (9.2)	36.1 (8.7)	0.830
*Civil status*			0.169
Married/cohabitant	7 (41.2)	11 (68.8)	
Divorced	2 (11.8)	0 (0)	
Single	8 (47.1)	5 (31.3)	
*Having children*			0.829
No	10 (58.8)	10 (62.5)	
Yes	7 (41.2)	6 (37.5)	
*Educational level*			0.909
PhD or Specialization	12 (70.6)	11 (68.8)	
University Degree	5 (29.4)	5 (31.3)	

**Table 2 ijerph-19-13788-t002:** Outcome variables of the two groups of participants to the trial.

Variables	Intervention Group Median (SD)	Control Group Median (SD)	*p*
MCS—pre	45.9 (33.2–55.7)	42.6 (25.1–54.7)	0.367
MCS—post	52.8 (42.5–57.6)	42.9 (29.2–56.6)	**0.006**
*p*	**0.002**	0.733	
PCS—pre	54.7 (42.4–58.8)	55.6 (44.8–62.6)	0.262
PCS—post	55.0 (43.7- 59.2)	56.7 (46.6–61.3)	0.203
*p*	0.653	0.397	
Positivity—pre	3.5 (3.0–4.5)	3.7 (2.7–4.7)	0.591
Positivity—post	3.4 (2.7–4.9)	3.5 (2.7–5.0)	0.986
*p*	0.373	0.627	
Job demand—pre	28.0 (20.0–37.0)	29 (24–35)	0.414
Job demand—post	28 (22–37)	29 (21–37)	0.790
*p*	0.231	0.555	
Decision latitude—pre	40.0 (38.0–48.0)	40.0 (32.0–44.0)	0.166
Decision latitude—post	42.0 (36.0–46.0)	40.0 (36.0–46.0)	0.260
*p*	0.340	0.135	

## Data Availability

The data presented in this study are available on request from the corresponding author.

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
