# Peer review of "Quality of Life and Stress Management in Healthcare Professionals of a Dental Care Setting at a Teaching Hospital in Rome: Results of a Randomized Controlled Clinical Trial"

_ijerph, 2022, doi:10.3390/ijerph192113788_

Round 1

Reviewer 1 Report (Previous Reviewer 2)

I thank the authors for improving the manuscript. It seems to me that the manuscript would benefit if the authors described in more detail the psychometric properties of the measures used (whether they were adapted into Italian, what Cronbach's alpha scores were found in the current study).

Author Response

response in attach

Reviewer 2 Report (Previous Reviewer 1)

Please, correct the following sentences:

- Short sets of 15 minutes were carried out developed across specific points

- A physical - respiratory - mental - behavioural form of training that teaches users to master the modulation of states of consciousness in order to influence body processes to wards greater health, wellbeing and better psychological/physiological conditions [2-26-27]. (no verb in the sentence)

- The main areas worked on shoulders, neck, chest, back, pelvis and the posterior muscles of the thigh (no verb in the sentence)

Author Response

response in attach

This manuscript is a resubmission of an earlier submission. The following is a list of the peer review reports and author responses from that submission.

Round 1

Reviewer 1 Report

I would like to thank the authors and editors for having had the opportunity to review this manuscript. The topic, managing occupational stress, is highly important. In this manuscript the authors seem to consider stress as a problem of an individual employee. However, psychosocial workload is an organizational issue, and it is the employer’s duty to assess it as all the other occupational safety and health risks at workplace and take preventive measures. This principle should not be forgotten.
-    I think the title should be shorter and more compact. You may consider leaving the name of the university away from the title.
-    Psychological stress of health care workers is a known problem and a lot of interventions have been performed, evaluated and published in the scientific literature. The authors should clearly point out, what new this study adds.
-    As the authors admit, the number of participants was very modest. Also, the duration of the intervention was quite short and the intensity relatively weak. The intervention seemed to be complex, multifaceted and the ultimate goal of it was not clear. I’m not convinced that the outcome measures are proper. SF12 is used as a questionnaire of life quality of chronic ill patients. Does it suit for evaluation of healthy workers? Why there are only two variables from the Karasek questionnaire (although the authors say they will use three: page 5, line 187). Job demand and Decision latitude are related to work organization and management. It is difficult to change them by personal interventions.
-    Page 2, line 55. This statement about burnout sounds very odd: “However, no cure has been indicated for the problem.” The authors refer here to the ICD 11 codes which don’t deal with treatment of any disease. There are plenty of studies about burnout and its treatment both at individual and organizational level, e.g. Salminen, 2021 (dissertation); Schaufeli & Enzmann, 1988; Kinnunen, Puolakanaho, Tolvanen, Mäkikangas, & Lappalainen, 2019; Awa, Plaumann, & Walter, 2010; Dreison et al., 2018. See also the systematic Cochrane review Preventing occupational stress in healthcare workers. I recommend being very precise in terminology. Burnout was not the topic of this study at all, in my opinion.
-    Page 2, line 58-59: Why is the dentists’ work considered as particularly challenging work? Compared to what? Please justify.
-    The content of the intervention should be described in plain language, no Indian yoga-terms, please.
-    Page 2, line 74: If the scope of the article is in quality of life and work-related stress, I think musculoskeletal problem should be excluded: “stress management techniques are effective in the prevention and management of musculoskeletal problems”
-    Figure 1: a box of the analysis of the intervention group is missing.
-    This is odd: Table 1: having sons. Do you mean children?
-    What is “degree” as an educational level?
-    Page 9 line 250-251: “We may also note as a point of both strength and weakness the unexpected physical activity carried out by the study participants.” I didn’t find any further information about the physical activity of the participants.

Author Response

I would like to thank the authors and editors for having had the opportunity to review this manuscript. The topic, managing occupational stress, is highly important. In this manuscript the authors seem to consider stress as a problem of an individual employee. However, psychosocial workload is an organizational issue, and it is the employer’s duty to assess it as all the other occupational safety and health risks at workplace and take preventive measures. This principle should not be forgotten.

attached the file with the answers

Reviewer 2 Report

Thank you for sharing this study. This is an interesting topic, but I had a few concerns that limited my enthusiasm for the manuscript in its current form:

1.       I suggest making the abstract clearer, adding information about the sample size and the conclusion of the study.

2.       The theoretical review too briefly summarizes previous research, although quality of life and stress management in health care workers have been fairly well studied. It might be useful to expand the introduction.

3.       I would like to see a more detailed description of the study procedure: inclusion and exclusion criteria, randomization procedure, etc.

4.       When describing the instruments, there is a lack of information about their psychometric properties. If these instruments have been adapted into Italian, it is important to cite the references. If they have not been adapted, the translation procedure should be described.

5.       In Table 2, the authors showed the mean values and standard deviations, although the Mann-Whitney test was used for comparison across groups. As far as I know, ranks are estimated for Mann-Whitney. I am curious to know the rationale for that decision.

6.       I suggest to add a section about the strengths and future directions to the paper.

Author Response

attached the file with the answers

Round 2

Reviewer 1 Report

Although the authors reply that they have deleted the parts that dealt only with burnout, burnout is mentioned several times throughout the article, e.g. in the abstract as an aim of the study to prevent burnout and on page 2 “Burnout syndromes are manifested in dealing

with problems related to the profession on a daily basis [5-6-7]”. This is confusing. The aims of the study are not clear. Exposure to physical, chemical, biological, ergonomic factors and musculoskeletal problems are also listed as problems of dental professionals. The authors should keep in the topic of the article. Burnout is certainly not the topic.

The authors conclude: “For these reasons, the results obtained cannot be generalized to the population from which they originate.”(Page9, lines 356-357). This is not understandable. If this is true, there is no reason to publish the article or claim that the study shows the effectiveness of the yoga intervention.

The language of the article is not the quality that I’m used to in scientific publications. You should be very precise what you say and concentrate in the outcomes and study questions that you should define unambiguously.